# Cardiovascular Toxicity of Antineoplastic Treatments in Hematological Diseases: Focus on Molecular Mechanisms to Improve Therapeutic Management

**DOI:** 10.3390/jcm13061574

**Published:** 2024-03-09

**Authors:** Serena Barachini, Gabriele Buda, Iacopo Petrini

**Affiliations:** 1Department of Clinical and Experimental Medicine, University of Pisa, 56126 Pisa, Italy; gabriele.buda@unipi.it; 2Department of Translational Research and New Technology in Medicine and Surgery, University of Pisa, 56126 Pisa, Italy; iacopo.petrini@unipi.it

**Keywords:** cardiovascular toxicity, hematological diseases, antineoplastic drugs

## Abstract

In recent years, advancements in the treatment of hematologic neoplasms have led to more effective and less toxic therapeutic schemes, resulting in prolonged patient life expectancy. However, the success of these treatments has also brought about an increased prevalence of cardiovascular adverse events, becoming a significant concern for the growing population of cancer survivors. Antineoplastic therapies, targeting both tumor and organ vessels, contribute to vascular toxicity, influenced by genetic factors and pre-existing vascular diseases. Chemotherapeutic agents and targeted treatments can induce cardiovascular toxicity by affecting endothelial cells and cardiomyocytes through various mechanisms, including hypoxia, vasculature abnormalities, and direct effects on cardiomyocytes. Cardiovascular adverse events encompass a wide range, from cardiac dysfunction to an elevated risk of arrhythmias. While early cardiac events are well-described in clinical trials, delayed toxicities are gaining relevance due to prolonged patient survival. The review focuses on the cardiac and vascular toxicity of antineoplastic drugs in hematological disorders, providing insights into the molecular physiopathology of cancer therapy-associated cardiotoxicity. Understanding how these drugs interact with the heart and blood vessels is essential for predicting, detecting, and managing chemotherapy-related heart issues.

## 1. Introduction

In recent years, there has been consistent progress in the treatment of neoplastic diseases. Therapeutic regimens are now more effective and less toxic, introducing new drugs that yield significant responses and extend patients’ life expectancy. However, the success in cancer treatment comes with the drawback of a broader spectrum of adverse events, particularly cardiovascular effects resulting from the treatment. The increasing number of cancer survivors undergoing treatment has raised concerns about these cardiovascular side effects [1,2].

Certain anticancer therapies target both tumor and organ vessels, contributing to the development of vascular toxicity, influenced by genetic factors and pre-existing vascular conditions. Additionally, some drugs can cause lasting structural and/or functional damage to cardiovascular structures. Adverse events are influenced by the pharmacodynamics and pharmacokinetics of the antineoplastic agent, and synergistic drug interactions in combination schedules can enhance these effects. Both traditional chemotherapeutic agents and targeted treatments can induce cardiovascular toxicity by impacting endothelial cells and cardiomyocytes through diverse mechanisms (Figure 1).

Cardiovascular adverse events encompass a wide range of conditions, including heart muscle dysfunction, an accelerated development of coronary artery disease, and an elevated risk of arrhythmias [3]. Early cardiac events have been well described in clinical trials, but delayed toxicities are gaining relevance due to the longer survival of patients. Sometimes, these effects can be underestimated in clinical trials given their delayed onset of presentation. These effects can be attributed to both the direct impact of therapeutic agents and the indirect consequences of accelerated heart disease in patients with pre-existing risk factors, heart conditions, and other health issues.

Cardiovascular complications arising from antineoplastic therapy can manifest as acute, subacute, or chronic and are subject to classification based on the nature of the damage, the specific organ structure or function affected, or the causative drug [4,5,6]. Two types of cardiotoxicities have been identified depending on the nature of the damage: Type I, involving irreversible damage to myocytes, leading to congestive heart failure and type II, representing a reversible decline in cardiac contractility, mainly caused by vascular endothelial growth factor inhibitors and tyrosine kinase inhibitors. There are numerous categories of chemotherapy agents, each of which induces side effects with varying incidence, intensity, and involvement of cardiovascular structures (Table 1). Cardiovascular adverse effects of antineoplastic drugs include not only left ventricular ejection fraction (LVEF) reduction and the development of heart failure (HF), but also cardiac dysfunction (cardiomyopathy, myocarditis), arrhythmias (QT prolongation, bradycardia or heart block, atrial arrhythmias, ventricular arrhythmias, or sudden death), vascular disease (ischemic vascular events, venous thromboembolism, pulmonary hypertension, hypertension), metabolic disorders (hyperlipidemia, impaired glucose tolerance), and pericardial disease [7].

Hypoxia, consequential vasculature abnormalities, and the direct effects of drugs on the mitochondria can lead to reduced ATP production in cardiomyocytes. This initially results in reversible cell injury, which may progress to irreversible injury and ultimately lead to cell death [8]. Moreover, the depletion of ATP or a direct effect of certain drugs can interfere with the functionality of ion exchange cell pumps on the cell surface, inducing electrical abnormalities that can cause arrhythmias [8]. Mitochondria make up around 30% of the volume of mature cardiomyocytes, supplying 95% of the energy needed for cardiac blood pumping. They are essential in myocardial cells for maintaining cellular homeostasis and are involved in regulating cellular metabolic signals such as calcium (Ca^2+^) homeostasis, reactive oxygen species (ROS) production, balancing redox state, autophagy, and apoptosis. Mitochondrial dysfunction is identified as one of the most responsible mechanisms for the cardiovascular toxicity induced by drugs [8,9,10]. Additionally, ROS induce the expression of VEGF in vascular endothelial and smooth muscle cells, further stimulating ROS production and increasing hypoxia-inducible transcription factors (HIF-1). Elevated ROS levels promote lipid peroxidation of unsaturated fatty acids, leading to apoptosis, autophagy, and ferroptosis [11,12]. The combination of nitric oxide with ROS forms reactive nitrogen species responsible for mitochondrial DNA damage in vascular endothelial cells, leading to hypertension [11].

**Table 1 jcm-13-01574-t001:** Summary of cardiovascular toxicities of novel antineoplastic drugs for hematological malignancies. Atrial fibrillation (AF), Bruton’s tyrosine kinase (BTK), bispecific antibodies (BsAbs), chimeric antigen receptor (CAR-T), cytokine release syndrome (CRS), left ventricular ejection fraction (LVEF), Janus kinases (JAK), histone deacetylase (HDAC), monoclonal antibodies (MoAbs), immunomodulatory (IMiDs), heart failure (HF).

Class of Drugs	Drug	Mechanism of Action	Adverse Event	Reference
Liposomal anthracyclines	Myocet (liposomal doxorubicin)	DNA intercalating agent, topoisomerase 2 inhibitor	Impairment of LVEF, HF	[13,14]
	Vyxeos (liposomal cytarabine plus daunorubicin)	DNA intercalating agent	QT interval prolongation and left ventricular dysfunction	[15]
BTK inhibitors	Ibrutinib	BTK	AF, ventricular tachycardia, ventricular fibrillation, sudden cardiac death	[16,17,18]
	Acalabrutinib (ACP-196)	BTK	Ischemic vascular events, AF, hypertension, ventricular arrhythmia, HF	[19]
	Zanubrutinib	BTK	AF, hypertension, ventricular arrhythmia	[16,18]
	Pirtobrutinib	BTK	AF	[20]
BCR-ABL inhibitors	Imatinib	BCR-ABL, PDGFR-α, and c-KIT	Cerebrovascular accidents	[21,22]
	Nilotinib	BCR-ABL, PDGFR-α, and c-KIT	Hypertension, ischemic vascular events, prolongation QT, hyperlipidemia, and hyperglycemia	[23,24]
	Bosutinib	BCR-ABL, FGFR 2, VEGFR 2, PDGFRβ, SRC	Ischemic vascular events, prolongation QT, hypertension, AF	[21,22]
	Ponatinib	ABL, FGFR 1,2 and 3, VEGFR 1, 2, and 3, PDGFR-α, PDGFR-β, c-KIT, SRC, TIE2	Hypertension, Ischemic vascular events, HF, pulmonary embolism	[25,26,27]
JAK inhibitors	Ruxolitinib	JAK/STAT	AF, dysregulation of lipolysis	[28,29,30]
PI3K inhibitors	Copanlisib	PI3K-α/PI3K-δ	Hypertension	[31,32,33,34]
HDAC inhibitors	Romidepsin	HDAC enzymes	Prolongation QT	[35,36,37]
	Belinostat	HDAC enzymes	Prolongation QT	[38]
MoAbs	Rituximab	Anti-CD-20	Infusion-related reactions, hypertension/hypotension, supraventricular tachycardia, AF	[39]
	Ofatumumab	Anti-CD-20	Infusion-related reactions, hypertension/hypotension, tachycardia	[40,41]
	Daratumumab	Anti-CD-38	Infusion-related reactions, AF, cardiac failure, coronary disease	[42]
	Brentuximab vedotin	Anti-CD-30	Sinus tachycardia hyperlipidemia, pulmonary embolism	[43,44,45]
	Obinutuzumab	Anti-CD-20	Infusion-related reactions, ischemic vascular events, AF, HF, pulmonary embolism	[46,47]
	Isatuximab	Anti-CD-38	Infusion-related reactions, AF,	[48]
BsAbs	Blinatumomab	CD-19/CD-3	CRS, tachycardia, AF, Ischemic vascular events, HF	[49,50,51]
	Teclistamab	BCMA/CD3	CRS, thrombocytopenia	[52]
	Talquetamab	GPRC5D/CD-3	CRS	[53]
	Mosunetuzumab	CD-20/CD-3	CRS	[49,54]
	Elranatamab	BCMA/CD-3	CRS, injection site reactions, upper respiratory tract infections	[55,56]
	Glofitamab	CD-20/CD-3	CRS	[49,57]
	Odronextamab	CD-20/CD-3	CRS	[49,58]
	Epcoritamab	CD-20/CD-3	CRS	[49,59]
	Alnuctamab	BCMA/CD3	CRS	[60]
CAR-T	Axicabtagene Ciloleucel	Anti-CD19 CAR T	CRS, atrial fibrillation, HF	[61,62]
	Tisagenlecleucel	Anti-CD19 CAR T	CRS, ventricular arrhythmias, AF, venous thromboembolic event	[63,64,65]
IMiDs	Lenalidomide	Cereblon protein	Ischemic vascular events, venous thromboembolic events, conduction abnormalities	[66]
	Pomalidomide	Cereblon protein	Venous thromboembolic events	[67]
Proteasome inhibitors	Bortezomib	Proteasome	Left ventricular dysfunction	[68,69]
	Carfilzomib	Proteasome	Hypertension, HF, arrhythmias, cardiomyopathy	[68,70,71]
Small molecule proteins	Venetoclax	Anti-BCL2	Hypertension, AF, cardiomyopathy, cardiac arrhythmia	[72,73,74]

Furthermore, the emergence of cardiac dysfunctions during cancer therapy can result in delays and dose reductions in antineoplastic agents, which could influence the overall cancer treatment outcomes. 

In this review, we focus on the cardiac and vascular toxicity of antineoplastic drugs in hematological disorders, describing progress in the knowledge of the molecular physiopathology of treatment-related adverse events. 

## 2. Anthracycline and Liposomal Anthracyclines

Anthracyclines such as doxorubicin (DOX), epirubicin, and daunorubicin are important chemotherapy drugs employed in the management of various cancer types, encompassing both solid tumors and non-solid malignancies, such as leukemia, lymphomas, carcinomas, and sarcomas. Anthracycline treatments significantly increase the risk of cardiomyopathy and HF in cancer survivors. This risk is directly related to the cumulative dose of anthracyclines, with a 25 times greater probability of developing cardiomyopathy in patients receiving 250 mg/m^2^ of DOX compared to those without exposure, with onset within 3.5 months and in 98% of cases within the first year after chemotherapy [75]. 

Anthracyclines are used in the treatment of approximately 50–70% of adult patients with lymphoma [76] and more than half of pediatric patients with cancer [77], although they represent the primary choice for the treatment of soft tissue sarcomas [78] and some breast cancers [79]. Anthracycline-related cardiotoxicity occurs in less than 5% of patients and manifests as electrocardiogram (ECG) changes (in 20–30% of patients) and arrhythmias (up to 3% of patients), including sinus tachycardia, supraventricular tachycardia, heart block, and ventricular arrhythmias. Symptoms range from palpitations, presyncope, and syncope to cardiac arrest, with some patients experiencing acute declines in cardiac function leading to dyspnea and HF. Pathologically, this condition resembles acute toxic myocarditis, featuring cardiomyocyte damage, inflammatory infiltrates, and interstitial edema [80]. Another retrospective study with over 4000 peditric solid tumor patients found clinically evident congestive heart failure in 2.2% of all anthracycline-treated patients [81]. While the risk of heart failure notably rises at cumulative doses exceeding 550 mg/m^2^ with DOX therapy, histologic alterations in endomyocardial biopsies have been observed in patients receiving only 240 mg/m^2^ [82]. Furthermore, Nousiainen et al. [83] demonstrated in lymphoma patients that a reduction in left ventricular ejection fraction exceeding 4% units following a cumulative DOX dose of 200 mg/m^2^ had a predictive accuracy with 90% sensitivity and 72% specificity for subsequent cardiotoxicity.

Topoisomerases, crucial DNA unwinding enzymes, are essential for cell replication playing a role in chromosome condensation, chromatid separation, and creating temporary DNA strand breaks. Topoisomerase 2β, primarily found in quiescent cells like cardiomyocytes, is associated with the nuclear matrix and the nucleolus. Anthracyclines target and inhibit topoisomerase 2β, forming persistent DNA cleavable complexes, causing DNA breaks, and ultimately leading to cancer cell death [84]. However, this inhibition can result in cardiotoxicity, potentially causing harm and cell death in cardiomyocytes. Furthermore, anthracyclines cause binding to mitochondrial DNA, resulting in reduced synthesis and subsequent damage to mitochondrial DNA. Oxidative stress is the result of either an excessive production of ROS that surpasses the capacity of the body’s oxidative defense mechanisms or the suppression of antioxidants. Anthracyclines impact the production of ROS through various mechanisms, which include their influence on the mitochondrial electron transport chain (NADH/Complex-1), the activity of nicotinamide adenine dinucleotide phosphate (NADPH) oxidases (NOX), and the malfunctioning of nitric oxide synthases.

To maintain robust ATP production, the heart has a higher concentration of mitochondria than other tissues. In cardiac myocytes, mitochondrial complex I, specifically associated with NADH dehydrogenase of the mitochondrial electron transport chain, in the presence of anthracycline, produces superoxide anions, increasing ROS production and mitochondrial dysfunction with decreased ATP synthesis and reduced calcium-uptake [85,86]. These events result in endoplasmic reticulum stress caused by high levels of cytoplasmatic calcium, which induce inflammation and cell death through mTOR inhibition. There are two types of endoplasmic reticulum stress: a beneficial stress that inhibits mTOR signaling and a detrimental stress that triggers cell death signaling. DOX induces endoplasmic reticulum stress and apoptosis in the heart, and the involvement of autophagy [87] adds complexity to its role [88]. DOX can up-regulate cardiac autophagy, but it can also inhibit lysosomal function, causing ROS accumulation. Mitophagy, the removal of damaged mitochondria, is impaired in doxorubicin-induced cardiotoxicity. Interventions like luteolin improve heart function by promoting mitochondrial autophagy in mouse cardiomyocytes [89].

Furthermore, anthracyclines are active on NOX, particularly NOX2, leading to an elevated generation of ROS [90]. The binding of anthracyclines to topoisomerase II within the nucleus can result in permanent DNA damage, and this event, together with oxidative stress, leads to p53 and nuclear factor-kB (NF-kB) activation. These molecular pathways contribute to heart dysfunction, cardiac fibrosis, apoptosis, necrosis, and inflammation.

Transient receptor potential (TRP) channels, sensitive to various factors including mechanical stress, play a role in pathological heart hypertrophy. The TRPC3-NOX2 complex stabilizes TRPC3, promoting harmful calcium entry and cardiac remodeling. In anthracycline-induced cardiac atrophy, higher TRPC3-NOX2 levels are linked to severity [91]. Inhibiting this complex reduces atrophy, while exercise, which induces physiological cardiac hypertrophy, lowers TRPC3-NOX2 levels. Nishiyama et al., showed that Ibudilast, an inhibitor of phosphodiesterase 4, disrupts the TRPC3-NOX2 complex, reducing doxorubicin-induced damage in mice [92].

DOX is a chemotherapy drug that causes cardiotoxicity, resulting in a severe and life-threatening heart condition known as DOX-induced cardiomyopathy (DIC), which is particularly lethal. Ferroptosis, whose endogenous regulator is glutathione peroxidase 4, is an iron-dependently regulated cell death because of lethal lipid oxidation. Tadokoro et al. [93], reported both a downregulation of glutathione peroxidase 4 and an induction of excessive lipid peroxidation through the DOX–Fe^2+^ complex in mitochondria. These effects lead to mitochondria-dependent ferroptosis as a potential molecular mechanism underlying DIC.

DOX treatment increases nitric oxide synthase expression in cardiac myocytes, leading to peroxynitrite production, which triggers cell death [94]. Endothelial nitric oxide synthase is also crucial in anthracycline-related endothelial damage. Endothelial nitric oxide synthase reduces DOX, generating oxidative and nitrosative stress [95]. These effects are most pronounced in endothelial mitochondria and are linked to increased ROS and reduced nitric oxide synthase and vascular endothelial growth factor (VEGF) levels [96].

Measures such as l-arginine treatment, mitochondrial antioxidants, and VEGF restoration can help reverse these effects and restore endothelial function [97,98]. Faridvand et al. [99] explored the protective effects of human amniotic membrane proteins (AMPs) against DOX-induced cardiotoxicity in rat cardiomyocyte cells. The authors showed that AMP pretreatment effectively reduced DOX-induced cell damage by lowering oxidative stress, maintaining mitochondrial health, and inhibiting apoptosis-related proteins suggesting AMPs as a potential therapeutic solution for DOX-induced heart damage.

Anthracycline-induced DNA damage and oxidative stress activate the p53 tumor suppressor. While deleting p53 in mice protects against anthracycline-induced early cardiac dysfunction, it increases the risk of late cardiotoxicity by blocking activation of the STAT3 transcription factor and leading to late-stage cardiac dysfunction [100,101].

ROS and nitrogen species can damage DNA, activating the nuclear enzyme poly (ADP-ribose) polymerase (PARP), which, when overactivated, depletes ATP, leading to cell dysfunction and necrosis [95]. PARP-1 also promotes the expression of pro-inflammatory genes, including NF-kB, whose prolonged activation in the heart triggers inflammation, endoplasmic reticulum stress, and cell death [102]. DOX increases NF-kB activity, potentially leading to cell apoptosis in cardiomyocytes and endothelial cells. Dihydrotanshinone I, a natural product, has shown enhanced heart function by suppressing the activation of macrophages and the excessive secretion of inflammatory cytokines, both in zebrafish and mice, demonstrating a cardio protective effect against DIC [103].

The most significant clinical impact of anthracycline cardiotoxicity is the reduction in size and mass of the left ventricular of the heart, cardiomyocyte atrophy, cardiac fibrosis and systolic cardiac dysfunction (acute and chronic) [104]. Furthermore, anthracycline chemotherapy has direct toxic effects on the vasculature, leading to a decreased ability to respond to substances that typically induce blood vessel dilation, such as acetylcholine [104].

Anthracycline-related heart problems are a major concern for cancer patients during and after treatment. These issues result from complex molecular mechanisms, with significant clinical implications. To address this, more research is needed to understand these mechanisms and develop innovative strategies for preventing and treating anthracycline-induced heart complications. One of the most promising techniques developed in recent years in the field of anthracyclines is the formulation of DOX molecules encapsulated within liposomes [105]. The encapsulation of a cytostatic agent in a macromolecular carrier, such as a liposome, drastically reduces its distribution volume, decreasing its diffusion in the body and consequently reducing toxicity to healthy tissues, while simultaneously increasing the drug concentration in neoplastic tissue. Under ideal conditions, the drug can be transported in the circulatory system within the aqueous space of the liposome and reach the tumor site in its active form. Encapsulation in a liposome means that the drug is protected from inactivation while in the bloodstream, and its diffusion through healthy endothelium is limited, while it can diffuse through tumor endothelium, which has fenestrations [106]. Myocet is DOX complexed with citrate ions, encapsulated in a non-pegylated liposome, a sterile and pyrogen-free aqueous dispersion of egg phosphatidylcholine and cholesterol that traps DOX by pushing it into the vesicle through the generation of a potential. Myocet^®^ demonstrates efficacy and safety both as a monotherapy and in combinations, leading to a decrease in the occurrence and intensity of cardiac events [107,108]. The incidence of hand-foot syndrome is extremely rare with Myocet, unlike pegylated liposomal DOX. Conjugation with liposomes allows for the use of much higher cumulative doses, enabling treatment even in patients who have already reached the maximum cumulative dose of DOX in previous therapies. Thanks to the significantly improved cardiac safety profile of liposomal DOX, it can also be used in patients with an increased risk of cardiac issues and can be administered along with new pharmacological molecules that carry varying degrees of cardiac risk [13,14,109].

CPX-351 (Vyxeos), a liposomal encapsulation of cytarabine and daunorubicin, was investigated in a phase 2 study on acute myeloid leukemia by Lin et al. [15] to assess its effect on cardiac repolarization. The authors showed that the metabolism and excretion of cytarabine and daunorubicin encapsulated in CPX-351 liposomes closely resembled that of their respective conventional formulations, and notably, CPX-351 did not acutely or sub-chronically prolong the QT/QTc interval, indicating a potential for reduced cardiotoxicity compared to conventional daunorubicin.

Various strategies can be employed to mitigate cardiovascular risks associated with anthracycline administration [110], including the use of a dose-reduced regimen, the utilization of liposomal DOX formulations to minimize myocardial drug accumulation, opting for infusional anthracycline administration, and considering concurrent administration of dexrazoxane. Dexrazoxane may offer cardioprotection through suggested mechanisms, particularly inhibiting the doxorubicin-topoisomerase-IIb interaction and chelating iron, thereby reducing cytosolic iron accumulation [110,111].

## 3. Alkylating and Platinum-Based Agents 

Hypertension, myocardial infarction, thromboembolism, and cerebrovascular disease are acute and late cardiovascular side effects associated with cisplatin. The exact mechanisms remain not fully understood but likely involve dysregulation of VEGF leading to endothelial dysfunction, vasodilation, capillary rarefaction, and oxidative stress [112]. Cisplatin-based chemotherapy carries a 9% risk of thromboembolic complications, likely due to endothelial damage and dysfunction, resulting in a hypercoagulable state. This can lead to cerebrovascular problems and contribute to atherosclerotic cardiovascular complications. Cisplatin-induced hypertension can cause acute cardiovascular issues during treatment [113]. Vascular toxicity resulting from chemotherapy often originates from impaired endothelial cell functions, which are responsible for sensing blood flow changes. Imbalances between the vasorelaxant nitric oxide and the vasoconstrictor endothelin-1 (ET-1) are a hallmark of endothelial dysfunction and contribute to the progression of vascular diseases. Initially asymptomatic, chemotherapy-induced endothelial cell dysfunction in the heart may, over time, increase the risk of cardiovascular conditions such as hypertension, coronary artery disease, and HF. Cisplatin is also associated with an elevation in Von Willebrand factor production, leading to endothelial activation [114]. The induction of cardiotoxicity by cisplatin involves the depletion of glutathione, a cellular antioxidant defense molecule, resulting in heightened cellular oxidative stress and adversely affecting mitochondrial respiration [115].

Alkylating agents exert antitumor activity either by cross-linking the two strands of DNA, preventing a complete separation (difunctional alkylators), or by binding to the DNA, causing DNA damage as cells attempt to remove this lesion (monofunctional alkylators). One of the most commonly used alkylating agents is cyclophosphamide, employed both in non-Hodgkin’s lymphoma, Burkitt’s lymphoma, multiple myeloma [116], and as an immunosuppressor in bone marrow transplantation. The pharmacological effect of cyclophosphamide varies with dosage; at low doses (<5 mg/kg), it acts as an immunosuppressor without known cardiotoxic effects, but higher doses (>200 mg/kg) used in treating hematological cancers and bone marrow transplantation show documented cardiovascular side effects. The incidence of cardiotoxicity ranges from 7% to 28%, with acute onset often within 21 days of therapy initiation [116]. Cyclophosphamide is metabolized by the liver into 4-hydroxy cyclophosphamide and subsequently reduced to aldophosphamide. This metabolite undergoes further metabolism to produce acrolein, which is implicated in the cardiotoxic effects of cyclophosphamide, contributing to enhanced oxidative and nitric stress in the heart [117]. Activation of the p53 and p38 mitogen-activated protein kinase pathways, induced by cyclophosphamide and its metabolites, results in cardiac apoptosis and hypertrophy. Furthermore, recent findings support a link between the use of cyclophosphamide and detrimental ventricular remodeling, characterized by compromised mitochondrial function and the initiation of endoplasmic reticulum stress [116].

Many patients undergoing alkylating agent treatment develop hypertension [11]. The pathophysiology appears linked to an imbalance in vasodilator/vasoconstrictor production, resulting in reduced nitric oxide, prostaglandin PGI2, and increased ET-1 and inducible nitric oxide synthase [116,118]. Specifically, cyclophosphamide reduces ATP production and fatty acid accumulation [119], leading to intracellular calcium buildup and increased pro-inflammatory cytokine levels, ultimately resulting in apoptosis due to mitochondrial damage from oxidative stress in endothelial cells [120].

Cyclophosphamide reduces free fatty acid (FFA) transport from the cytosol to the mitochondria, decreasing beta-oxidation and ATP production in cardiomyocytes. Oxidative stress leads to intracellular calcium accumulation, activating apoptosis and inflammation through various pathways, including toll-like receptor-4 (TRL-4), NF-kB, tumor growth factor-beta (TGF-β), and tumor necrosis factor-alpha (TNF-α), ultimately causing cardiac hypertrophy and cardiac remodeling [121]. The TLR4/NF-kB signaling pathway significantly contributes to the onset and progression of inflammation and oxidative stress. Activated by lipopolysaccharide (LPS) and ROS, this pathway stimulates the expression of inflammatory cytokines such as IL-β and TNF-α, leading to inflammatory cell infiltration and myocardial fibrosis. Moreover, TLR4/NF-kB activation also triggers the expression of ROS, generated by the mitochondrial electron transport chain, NADPH oxidase, and xanthine oxidase, leading to the damage of the cardiac and vascular endothelium, along with the exacerbation of myocardial hypoxia [122]. Recent studies have indicated that the primary mechanisms by which TLR4/NF-kB exacerbates myocardial infarction encompass inflammation [123], oxidative stress [124], and pyroptosis—an inflammatory form of cell death that governs cardiomyocyte loss post-myocardial infarction—as well as apoptosis [125,126].

## 4. Tyrosine Kinase Inhibitors

The phosphorylation of tyrosine residues increases in cells following neoplastic transformation. Mutations in tyrosine kinases can constitutively activate these enzymes, leading to the transformation of cells. Mutations in multiple tyrosine kinases have been described in various hematological malignancies, and specific inhibitors have been developed with astonishing clinical results [127].

### 4.1. BCR-ABL Inhibitors

In chronic myelogenous leukemia (CML), the Abelson 1 kinase (ABL1) undergoes constitutive activation due to the reciprocal translocation t (9:22) (known as the Philadelphia chromosome, Ph). This translocation juxtaposes the breakpoint cluster region (BCR) gene with the ABL kinase gene, giving rise to the formation of the deregulated BCR-ABL1 kinase. BCR-ABL1 resulted in the constitutive activation of a plethora of signaling pathways, including PI3K/AKT/mTOR, RAS/RAF/MAPK, JAK-STAT, and WNT/β-catenin, that regulate cell survival, disease progression, and genomic stability. Inhibitors targeting the ABL1 kinase have demonstrated remarkable effectiveness in the treatment of CML. The introduction of targeted therapy with BCR-ABL tyrosine kinase inhibitors (TKIs) has transformed the management of patients with CML [128].

Kinases play a crucial role in maintaining the balance of cardiac, vascular, and metabolic functions. Any alteration in kinase activity by TKI may have not only an oncogenic role but also diverse effects on the vasculature. This “on-target” toxicity contrasts with “off-target” toxicity, where the non-specific nature of TKIs may inhibit structurally similar or related kinase receptors, potentially causing vascular diseases [129]. Nevertheless, the cardiotoxic effects linked to BCR-ABL1 TKIs can lead to direct or indirect impairment of mitochondria, including changes in the ROS/redox system, mitochondrial Ca^2+^ homeostasis system imbalance, and ER stress signaling [10].

TKIs are associated with various cardiotoxicities including hypertension, ischemic heart disease, cerebrovascular disease, peripheral arterial disease, arrhythmia, myocardial fibrosis, acute coronary syndromes, and HF [130,131]. The mechanisms contributing to arterial hypertension encompass both functional irregularities (deactivation of endothelial nitric oxide synthase and the release of vasoconstrictors such as ET-1) and structural abnormalities (capillary rarefaction resulting from pericytes loss due to PDGFR inhibition and the hindrance of angiogenesis). These effects may be exacerbated by concurrent renal dysfunction. 

Cardiovascular toxicity varies significantly among different TKIs, and it is linked to the number of kinases inhibited. Notably, TKIs affecting the VEGF and MAPK/ERK kinase pathways pose the highest clinical risk of cardiotoxicity [132]. Nilotinib, dasatinib, and imatinib are employed in newly diagnosed Ph chromosome-positive CML. Dasatinib and imatinib are also used for acute lymphoblastic leukemia. Nilotinib and dasatinib are associated with HF and arrhythmia, including QT prolongation, with an average increase of 15 milliseconds from baseline [80].

Elmadani et al. [133] demonstrated that treating neonatal rat ventricular cardiomyocytes with dasatinib reduces viability and inhibits ERK phosphorylation. Additionally, they showed that dasatinib induces dose-dependent endothelial cell death, potentially increasing the risk of pulmonary hypertension. Disturbances in endothelial cell homeostasis may worsen dasatinib’s cardiotoxic effects on adjacent cardiomyocytes. Furthermore, Yue et al. [134] showed that inhibiting ERK, regulated by c-Src, enhances ischemia/reperfusion-induced apoptosis in neonatal cardiac myocytes exposed to ischemia in rats.

Imatinib induces cell death in cardiomyocytes by altering cardiac cell homeostasis via the induction of the endoplasmic reticulum stress response, ROS production, and activation of catabolic hydrolases such as caspase [135]. Cardiotoxic imatinib-induced events include HF and myocardial infarction with cardiogenic shock [10,136].

Nilotinib and bosutinib cause mitochondrial damage by indirectly reducing mitochondrial ATP, impairing respiratory chain function, altering mitochondrial permeability, and inducing endoplasmic reticulum stress. The 6-year follow-up safety data from ENESTnd nilotinib demonstrated more cardiovascular events compared with imatinib [23]. Nilotinib has been found to increase the expression of pro-atherogenic adhesion proteins on human endothelial cells, such as intercellular adhesion molecule 1 (ICAM1), vascular cell adhesion protein 1 (VCAM1), and E-selectin, promoting vascular events. The upregulation of adhesion molecules is linked to a reduction in the level of miR-3121–3p, leading to the additional upregulation of IL-1β. It is suggested that targeting the miR-3121–3p/IL-1β axis could be a potential strategy to prevent vascular events in CML patients identified as high risk [137].

CML patients treated with nilotinib demonstrated increased platelet adhesion and elevated expression of soluble P- and E-selectin, soluble intercellular adhesion molecule-1 (sICAM-1), soluble vascular cell adhesion molecule-1 (sVCAM-1), TNF-α, IL-6 levels, and endogenous thrombin potential [138].

Treatment of atherogenic mice with nilotinib resulted in increased atherosclerotic buildup and blocked reperfusion and angiogenesis in a hind-limb-ischemia model of arterial occlusion. In mouse model studies, nilotinib significantly enhanced thrombus growth and stability in damaged mesenteric arterioles and the carotid artery [139]. Treatment with nilotinib in a mouse model increased the growth and stability of thrombi, blocking reperfusion and angiogenesis and developing atherosclerotic plaques. 

Brümmendorf et al. [21] compared the safety and efficacy of bosutinib versus imatinib in a phase 3 BFORE trial enrolling CML patients. The authors showed that bosutinib adverse events included acute cardiac failure, myocardial ischemia, and renal failure. 

Ponatinib, a third-generation TKI, not only induces pro-thrombotic effects [140], enhancing platelet activation and adhesion, but also induces hypertension [141,142]. Ponatinib is associated with a higher risk of cardiovascular side effects compared to other TKIs [25]. In the phase 2 clinical trial (PACE), ponatinib showed a considerable response in CML patients but caused different adverse effects, including hypertension, coronary, cerebrovascular, and peripheral vascular events, especially in patients with pre-existing cardiovascular risk factors [25]. The exact mechanism behind ponatinib-induced adverse vascular effects is unclear, but it is attributed to the drug’s broad kinase inhibition profile. Ponatinib is thought to cause endothelial dysfunction by non-specifically targeting vascular endothelial growth factor receptors. Additionally, it increases the risk of vascular occlusive events by promoting the expression of proatherogenic surface adhesion receptors and has direct prothrombotic effects by accelerating platelet activation and adhesion [143].

Madonna et al. [144] in an in vitro model, demonstrated that ponatinib inhibits endothelial survival, reduces angiogenesis, and induces endothelial senescence and apoptosis via the Notch-1 pathway. Selective blockade of Notch-1 was effective in preventing ponatinib-induced vascular toxicity. In addition, the same authors showed that empagliflozin and dapagliflozin, two anti-diabetic drugs that inhibit sodium glucose co-transporter-2, attenuate the vasculo-toxic effect exerted by ponatinib by reverting endothelial cell senescence and dysfunction [145].

Among all FDA-approved TKIs, ponatinib exhibits more cardiotoxicity [146]. Only ponatinib is active in CML patients, whose acquired resistance is determined by the T315I mutation. The reduction in ponatinib dose has reduced the risk of cardiac adverse events, preserving good antitumoral efficacy [147].

BCR-ABL1 TKIs induce endoplasmic reticulum stress, activating PRK-like endoplasmic reticulum kinase and IRE1 pathways, leading to pro-apoptotic BAX protein release, mitochondrial depolarization, ATP depletion, and cytochrome c release. Endoplasmic reticulum stress induces Ca^2+^ release into mitochondria, opening the mitochondrial permeability transition pore, causing disruptions in mitochondrial membrane potential, releasing cytochrome c, activating caspase, and inducing apoptosis. Dysfunctional mitochondria affect mitochondrial oxidative phosphorylation and the tricarboxylic acid cycle and exhibit an uncoupled electron transport chain increasing ROS production, further damaging mitochondria [10].

Understanding the mitochondrial processes implicated in BCR-ABL1 TKI-induced cardiotoxicity is crucial for formulating effective strategies to avert cardiomyocyte dysfunction in individuals with CML.

### 4.2. Bruton’s Tyrosine Kinase Protein Inhibitors

In the last ten years, the chronic lymphocytic leukemia (CLL) treatment landscape has undergone a significant transformation, transitioning from cytotoxic chemotherapy to targeted therapies [148]. The advent of Bruton’s tyrosine kinase inhibitors (BTKIs) has revolutionized CLL treatment and found application in various other malignancies. BTK is a member of the Tec kinase family and is a key player in B-cell activation and proliferation BTK inhibition has become a crucial strategy to disrupt the unchecked signaling that drives B-cell proliferation and contributes to various B-cell malignancies, including CLL.

The toxicities associated with ibrutinib, a BTK inhibitor, include atrial fibrillation in up to 38% of patients [149], ventricular arrhythmias, HF, and hypertension [20]. Recent evidence indicates that newer BTK inhibitors, such as acalabrutinib and zanubrutinib, have less cardiac activity and are more selective for BTK [16].

In the long-term follow-up of the phase III ASPEN study, zanubrutinib demonstrated a lower incidence of atrial fibrillation/flutter (23.5% with ibrutinib versus 7.9% with zanubrutinib) and hypertension (25.5% with ibrutinib versus 14.9% with zanubrutinib) in patients with Waldenström macroglobulinemia [150].

The mechanisms of atrial fibrillation associated with ibrutinib and other BTK inhibitors involve the inhibition of C-terminal Src kinase (CSK), enriched in the atria, as well as the inhibition of the PI3K/AKT and ion channel pathways, potentially enhancing Ca^2+^/calmodulin-dependent protein kinase II (CaMKII) [151]. Deletion of CKS has been associated with an increase in the incidence of atrial fibrillation, cardiac fibrosis, and inflammation (IL-6 mediated) [152]. Notably, off-target kinase inhibition causing the suppression of PI3K-AKT can result in disorganization of myocardial cells, fibrosis, disturbances in calcium signaling, and ultimately, culminate in HF [20].

BTK inhibitors have the potential to interfere with PI3K pathways in cardiomyocytes, affecting normal ion currents by activating late sodium current, which leads to action potential prolongation and abnormal automaticity. This ultimately causes arrhythmias such as AF and QT interval prolongation. In platelets, activation of BTK occurs through the binding of Von-Willebrand factor, collagen, and fibrinogen to cognate glycoproteins, leading to platelet activation [153,154]. In addition, in vitro studies suggest that the impairment of insulin-like growth factor 1-dependent activation of intracellular calcium handling plays a crucial role in the susceptibility to ventricular arrhythmias with both ibrutinib and acalabrutinib [155].

Various potential mechanisms have been suggested to play a role in the exacerbation or onset of hypertension linked to BTK inhibitors, specifically ibrutinib and acalabrutinib: inhibition of the PI3K pathway leading to vascular tissue fibrosis [156] and down-regulation of nitric oxide formation in bone marrow-derived dendritic cells, resulting in dysregulation of vascular tone [157].

In addition to atrial fibrillation, stroke is a major adverse event associated with next-generation BTK inhibitors, including acalabrutinib or zanubrutinib. Probably, the risk is increased by the additive effect of atrial fibrillation and inherent proinflammatory states.

In conclusion, BTK inhibitor therapies have transformed the management of CLL and various hematological malignancies. However, even with the advent of next-generation BTK inhibitors, the occurrence of cardiotoxic events remains a potential limitation to the optimal utilization of these treatments. The medication management for these patients should include a multidisciplinary strategy involving the control of hypertension, heart rate, and fluid status, as well as electrocardiogram and cardiac imaging data [158].

#### 4.2.1. Janus Kinase Inhibitors

Janus kinases (JAKs) are a family of intracellular tyrosine kinase proteins associated with transmembrane receptors. The binding of the receptors with their ligands, including growth factors, cytokines, and hormones, induces the transphosphorylation of JAKs and the activation of signal transduction. JAKs play a significant role in myeloproliferative disorders (MPNs), where heightened activation of the JAK/STAT pathways in hematopoietic stem cells leads to uncontrolled proliferation and cytokine production [159].

In 2011, the first JAK1/2 inhibitor, ruxolitinib, was approved for the treatment of primary and secondary myelofibrosis [28]. Subsequently, its approval was extended to include polycythemia vera and acute graft versus host disease. The COMFORT-1 and COMFORT-2 studies compared ruxolitinib with placebo and the best available therapy, respectively, for myelofibrosis treatment, showing a higher proportion of patients achieving a 35% or greater reduction in spleen volume with ruxolitinib, irrespective of JAK2 mutation status [160,161,162]. The cardiotoxic effects associated with Ruxolitinib are not fully understood; nevertheless, arterial hypertension could be a common comorbidity, as there may be a notable increase in systolic blood pressure after 72 weeks of treatment with no significant alterations in diastolic blood pressure. Patients may manifest a deterioration in pre-existing hypertension or a new-onset disease [163]. In particular, Ruxolitinib may contribute to the development of hypertension by inhibiting the JAK/STAT signaling pathway in adipose tissue, promoting weight gain, and interfering with growth hormone by STAT5 phosphorylation and leptin pathways in adipocytes [154,163]. Inhibition of JAK/STAT signaling in adipocytes leads to dysregulation of lipolysis, metabolism, and blood pressure homeostasis [164]. Jak2 is a key regulator of cytokine and growth factor stimulation, leading to downstream activation of Stat3 signaling and also activation of the non-canonical PI3K and MAPK cascades, all of which are involved in the pathogenesis of idiopathic pulmonary fibrosis [165]. In pulmonary hypertension, the process of structural remodeling in the pulmonary vascular system, involving both smooth muscle cells and endothelial cells of the pulmonary arteries, as well as immune system cells, leads to luminal obstruction and a consequent increase in pulmonary arterial pressure with right ventricular hypertrophy. If left untreated, this condition can result in death due to HF [165]. No significant electrocardiographic changes have been reported, but a case of pulmonary hypertension with left ventricular dysfunction has been noted after two treatment regimens (14 months total) of treatment with ruxolitinib in a 57-year-old female, with myelofibrosis [29,30,166].

Sapre et al. [30] performed a retrospective study on myeloproliferative neoplasms patients initiating ruxolitinib treatment, revealing elevated systolic blood pressure at 72 weeks. In a parallel mouse model, ruxolitinib was found to suppress JAK/STAT signaling in adipose tissue, potentially leading to weight gain. Previously established as a pivotal regulator of adipose tissue lipolysis, JAK/STAT signaling [167], coupled with the known activation of lipolytic signaling by growth hormone in adipose tissue, the authors suggest that the inhibition of JAK/STAT could contribute to heightened adipose tissue accumulation. With the development and utilization of pharmaceutical JAK1/2 inhibitors in clinical settings, it becomes imperative to comprehend the enduring metabolic repercussions associated with their prolonged use.

#### 4.2.2. PI3K Inhibitors

The PI3K/Akt/mTOR signaling pathway is a crucial intracellular signaling cascade that regulates several metabolic processes, including an increase in cell size, survival, proliferation, and insulin and glucose metabolism [168]. Hyperactivation of the PI3K/Akt/mTOR signaling pathway is commonly observed in different tumors, and key components within this pathway are often dysregulated in various types of cancer. Several categories of PI3K inhibitors have been developed, including pan-class I PI3K inhibitors that hinder the activation of all four PI3K class-I isoforms (α, β, γ, and δ) as well as isoform-specific PI3K inhibitors.

The PI3Kδ specific inhibitor idelalisib is the first FDA approved PI3K inhibitor to treat relapsed CLL, follicular B-cell non-Hodgkin lymphoma, and small lymphocytic lymphoma [169]. Several second-generation PI3K inhibitors have been formulated and are presently undergoing clinical assessment including isoform-specific inhibitors characterized by enhanced selectivity, inhibitors designed to target multiple PI3K isoforms with a synergistic impact (duvelisib, a dual inhibitor of PI3Kδ and γ), and pan-class I inhibitors (Copanlisib) [170].

PI3K/AKT/mTOR signaling, which is a key player in proliferation, cell survival, and angiogenesis, is frequently deregulated in malignant lymphoma, leading to uncontrolled proliferation of B cells. The majority of these small molecule inhibitors targeting PI3K are ATP-competitive kinase inhibitors. The various isoforms of PI3K participate in tissue-specific signaling pathways, while PI3Kδ and PI3Kγ are predominantly found in hematopoietic cells, PI3Kα and β are universally expressed. PI3Kα has been demonstrated to play a role in glucose metabolism through the insulin growth factor receptor. PI3Kβ is essential for platelet adhesion and aggregation, and PI3Kδ is activated through the B-cell receptor, transmitting signals for survival and proliferation through the NF-kB pathway, a critical process for the development and expansion of both normal and malignant B-cells. In contrast, PI3Kγ is present in tumor-associated macrophages and contributes to tumor-related immune suppression [171]. Idelalisib is a potent PI3Kδ inhibitor, while copanlisib is a pan-specific PI3K small molecule inhibitor for four key isoforms with increased activity against PI3Kα and PI3Kδ [171].

Dreyling et al. [32] evaluated the efficacy and safety of single-agent copanlisib in patients with histologically confirmed indolent B-cell lymphoma who had relapsed after or were refractory to at least two prior lines of treatment in a large phase 2, multicenter, open-label study (CHRONOS-1). The authors showed that copanlisib is safe and well tolerated, and the most frequent adverse events observed were hyperglycemia and hypertension, both transient and manageable [31,32]. Similarly, copanlisib has demonstrated hypertension and hyperglycemia as adverse effects, in a phase II study in patients with diffuse large B-cell lymphoma [33].

Recommendations for management of these patients are glucose-lowering medications and short-acting anti-hypertensive therapy. Because PI3K inhibitors may increase the susceptibility to opportunistic infections, it is advisable to implement prophylactic measures against pneumocystis and employ preventive strategies for cytomegalovirus [172].

## 5. BCL2 Inhibitors

Venetoclax is a promising agent that has demonstrated high efficacy in various hematological diseases, particularly CLL, acute myeloid leukemia (AML), and multiple myeloma [173]. The B-cell leukemia/lymphoma-2 (Bcl-2) family proteins exert their anti-apoptotic effect by inhibiting BAX, BAK, and BH3-only proteins [174] that induce permeabilization of the mitochondrial outer membrane and release of cytochrome c, activating caspases and triggering cell death [173].

BCL-2 overexpression leads to chemoresistance [175,176]. Venetoclax, a potent and selective BH3-mimetic, targets the BH3 domain of BCL-2, restoring apoptosis in tumor cells [72,177,178]. Davids et al. [179] conducted an integrated safety analysis of venetoclax monotherapy in 350 patients with CLL from three phase I/II studies, revealing that venetoclax treatment may induce cardiac toxic effects, such as hypertension, predominantly of grade 1/2 severity, in 6% of patients with first onset after 1 year on therapy.

In other clinical contexts, venetoclax treatment can cause toxic effects on the heart, such as cardiomyopathy and cardiac arrhythmia [179] and can induce apoptosis in other organs, leading to organ toxicity such as acute kidney injury and tumor lysis syndrome [180,181]. AlAsmari et al. showed the effects of venetoclax on the heart in rats, demonstrating toxic effects on cardiomyocytes, including an increase in cardiac enzymes, such as myocardial muscle creatine kinase (CK-MB) and cardiac troponin (cTn-I). Additionally, there were alterations in the expression of relevant genes associated with cardiac injury, induction of apoptosis, and changes in oxidative stress mediated by the upregulation of interferon-gamma (INF-δ), TGF-β, and NF-kB [73]. Furthermore, there was an increase in the protein expression of IL6 and TNF-α, while the antioxidant superoxide dismutase-2 showed a decrease. It’s important to note that these findings were observed with very high doses of Venetoclax in rats, and the impact of the drug at therapeutic doses in humans remains uncertain [73].

## 6. Histone Deacetylase Inhibitors

Histone deacetylase (HDAC) inhibitors are a group of anticancer agents that modify the post-transcriptional activity of proteins by inactivating histone deacetylase enzymes, thereby leading to cell cycle arrest by increasing cyclin-dependent kinase inhibitor 1 (p21) levels, and cell death or apoptosis of cancer cells by activation of p53 [182,183].

HDACs can increase the expression of p21waf1 and reduce levels of cyclin A and D, as well as nitric oxide, TNF-α, IFN-δ, IL-6, and IL-12. Galimberti et al. [184] demonstrated that HDACs induce cytotoxic and pro-apoptotic effects in acute myeloid leukemia HL-60 cells by reducing BCL-2 and increasing BAK protein levels, resulting in mitochondrial membrane depolarization and the release of cytochrome c. Subsequently, this process promotes the activation of caspase-9 and induces apoptosis.

Vinodhkumar et al. [185] illustrated the mechanism associated with the anticancer effects of romidepsin on A549 lung cancer cells. They demonstrated the inhibition of cell proliferation through apoptosis and G2/M phase cell cycle arrest, down-regulating proteins such as phosphorylated pRb, cyclin B1, Cdc2/Cdk-1, and up-regulating p21 expression. Panicker et al. [186] evaluated the anticancer activity of romidepsin both in vitro and in vivo, demonstrating the inhibition of cell proliferation with caspase-dependent apoptosis in vitro and tumor growth inhibition in vivo. Furthermore, this HADC inhibitor induced gene expressions of neurotrophin receptor p75, neurotrophic tyrosine kinase receptor type 1, and p21. Similarly, Li et al. [187] demonstrated that romidepsin inhibits the proliferation of hepatocarcinoma cell lines and induces apoptosis through the activation of caspase/PARP, mediated by the induction of p53/p21 signaling pathways. Romidepsin initiates apoptosis in myeloid leukemia cells predominantly via the mitochondrial pathway, induced by the translocation of Bax, a critical factor in determining cell fate. The activation of caspase-9 and the ensuing cascade is triggered by romidepsin, causing perturbation of the mitochondrial membrane, release of cytochrome c, and resultant mitochondrial damage [188].

The HDAC inhibitors are approved to treat cutaneous or peripheral T-cell lymphoma. Several HDAC inhibitors have demonstrated QT interval prolongation. The data obtained from the cardiac monitoring of 42 patients treated on a phase II trial with Romidepsin showed that this drug is associated with QT prolongation [189,190]. Similarly, results of a phase II clinical trial (BELIEF) involving ECG analysis of 129 patients treated with belinostat, identified two patients with grade 3 QT prolongation [38].

HDAC inhibitors have been associated with delayed cardiac repolarization and in rare instances, a lethal ventricular tachyarrhythmia known as torsades de pointes. The mechanism(s) of HDAC inhibitor-induced effects on cardiac repolarization is unknown. The prolongation of cardiac repolarization may be mediated in part by transcriptional changes of genes required for ion channel trafficking and localization to the sarcolemma [190,191,192].

The mechanisms of HDAC inhibitor-induced effects on cardiac repolarization are not well elucidated, yet. However, transcriptional profiling of ventricular myocardium from dogs treated with HDAC inhibitors suggests that corrected QT interval (QTc) prolongation can be due to alterations in the transcription of genes necessary for the trafficking and localization of HERG-K+ ion channels to the sarcolemma [191]. The risk of prolonged QTc intervals is associated with the impact of anticancer drugs on both the increase in inward current and the decrease in outward current. This leads to an extension of the ventricular action potential, particularly during the repolarization phase. Repolarization is governed by two types of delayed rectifier K+ currents, namely IKr (rapid) and IKs (slow). The majority of drug-induced QTc prolongations are linked to the inhibition of IKr, a current carried by the potassium voltage-gated channel subfamily H member 2, commonly referred to as the hERG channel [80].

HDAC inhibitors, such as romidepsin, exhibit a diverse cardiac safety profile. Initial trials indicated an elevated risk of cardiac events, but subsequent adjustments in enrolment criteria mitigated this risk [193,194].

Caution is recommended for patients with pre-existing cardiac conditions, requiring regular electrolyte monitoring. Awareness among oncologists and cardiologists about HDAC inhibitor-induced ECG changes is crucial.

## 7. Immunomodulatory Drugs

The therapeutic approach of targeted protein degradation has experienced notable advancement and considerable investment in recent years. The immunomodulatory (IMiDs) drugs thalidomide, lenalidomide, or pomalidomide have been approved in the clinic for the treatment of multiple myeloma and myelodysplastic syndrome (MDS) with deletion of chromosome 5q [195].

Cereblon (CRBN) is a substrate recognition protein in the E3-ligase ubiquitin complex, and IMiDS acts via the CUL4-RBX1-DDB1-CRBN (CRL4CRBN) E3 ligase by acting as a molecular glue degrader to scaffold protein-protein interactions [195]. IMiDs have been linked to cardiotoxicity, alongside their widely recognized heightened risk of vascular complications such as venous thromboembolism. The precise mechanism underlying this cardiotoxicity is still not fully understood. However, there are indications that it may involve proteasome-mediated protein degradation associated with the binding to cereblon resulting in its activation of the E3 ubiquitin ligase. Additionally, endothelial injury and dysfunction have been proposed to have a significant role in this process [196]. In particular, IMiDs have the potential to generate an imbalance between procoagulant and anticoagulant proteins on endothelial cell surfaces leading to an increase in phosphatidylserine and tissue factor expression, activation of glycoprotein GPIIb/IIIa and suppression of endothelial protein C receptor and thrombomodulin expression. IMiDs may also induce an increase in the platelet activator cathepsin G, factor VIII, and von Willebrand factor, inhibit cyclooxygenase-2, and reduce prostaglandin E2 production, leading to endothelial cell stress and injury [197].

Higher doses of thalidomide can be associated with bradycardia and atrioventricular conduction abnormalities [198].

In the study of Das et al. [199], the authors showed an increased risk of high-grade cardiotoxic events (grade ≥ 3) in multiple myeloma patients treated with IMiDs (thalidomide or lenalidomide), compared to those not receiving IMiDs, even though a potential bias might have been due to the high dose of DOX used in combination with IMiDs. Dimopoulos et al., presented data from two phase III studies involving patients with relapsed disease, where lenalidomide, when combined with dexamethasone, has been linked to an elevated risk of myocardial infarction and stroke compared to dexamethasone alone [200]. In addition, there have been two case reports of fatal myocarditis possibly related to lenalidomide [201,202].

Data from the Food and Drug Administration (FDA) Adverse Events Reporting System (FAERS) database reveal elevated odds ratios (ORs) for adverse events (AEs) associated with thalidomide, encompassing atrial fibrillation, cardiac failure, and coronary artery disease. Although FAERS data rely on voluntary reporting and may not establish causation, it remains a standard reference for toxicity profiles.

Lenalidomide exhibited increased ORs for atrial fibrillation, cardiac failure, and coronary artery disease, when compared to alternative agents [42].

Recently, Mateos et al. [203] and Matsumoto et al. [204], presented data from a phase III trial (KEYNOTE-183) in pretreated patients with multiple myeloma, which compared pomalidomide combined with dexamethasone with or without the addition of pembrolizumab. In this trial, deaths related to myocardial infarction, cardiac failure, pericardial hemorrhage, and myocarditis have been reported. The addiction to anti PD1 antibodies increases the risk of immune related adverse events, including myocarditis. Myocarditis can occur during treatment with immune checkpoint inhibitors and can lead to patient death if not promptly treated. Fortunately, myocarditis is not common in cancer patients treated with immune check point inhibitors, with the possible exception of patients harboring thymic carcinomas, for which special attention is recommended [205,206].

## 8. Proteasome Inhibitors

High grade cardiac toxicity has been reported with proteasome inhibitors such as Bortezomib and Carfilzomib (OR 1.67–2.68 depending on the specific agent, and generally higher with carfilzomib) [1].

The proposed underlying mechanism of cardiac toxicity associated with bortezomib may be linked to the hindrance of NF-kB activation in cardiomyocytes [207]. This interference impacts angiogenesis and the survival of cardiac myocytes, leading to protein accumulation and mitochondrial dysfunction, thereby affecting contractility [208,209]. In fact, blocking the proteasomal-dependent degradation of sarcomere proteins may result in the abnormal buildup of ubiquitinated proteins that interact to create progressively more advanced protein aggregates detrimental to cellular function. This, in turn, contributes to heightened mitochondrial dysfunction and stress in the sarcoplasmic/endoplasmic reticulum [207].

Furthermore, additional studies have reported that bortezomib may concurrently contribute to the progression of atherosclerotic plaques and a tendency to rupture, while also facilitating ischemic heart complications by reducing or abrogating myocardial preconditioning [210]. Cardiovascular complications associated with bortezomib treatment encompass various adverse events, including atrioventricular block, arrhythmias, fibrillation, ischemic heart disease, pericardial effusion, and orthostatic hypotension [211].

Carfilzomib treatment reduces the mitochondrial membrane potential, ATP production, and mitochondrial oxidative respiration, increasing mitochondrial oxidative stress, leading to a decrease in contractility of cardiomyocytes. In addition, it downregulates the expression of genes involved in extracellular matrices, the integrin complex, cardiac contraction, as well as autophagy, and upregulates stress responsive proteins, including heat shock proteins [207,212]. Moreover, patients who have cardiovascular adverse events with carfilzomib, exhibited a down-regulation of pyruvate and the up-regulation of lactate dehydrogenase B, suggesting a potential role of the pyruvate oxidation pathway in mitochondrial dysfunction [213]. Carfilzomib also influences vascular smooth muscle cells, potentially exacerbating the vulnerability of atherosclerotic plaques [214]. Versari et al. [215] showed increased ubiquitin conjugates in smooth muscle cells and macrophages, which may become cytotoxic and lead to heightened oxidative stress and apoptosis. This could contribute to fibrous cap weakening and necrotic core enlargement. Impaired proteasome-dependent degradation may worsen damage, fostering plaque instability through the accumulation of oxidized and ubiquitinated proteins [215].

The combinations of the chemotherapy drugs need to be carefully considered in patients with multiple myeloma and depend on at least doublet, triplet, or more recently quadruplet regimens, due to the increase in risk of cardiac toxicities, including arrhythmias and QT prolongation.

## 9. Monoclonal Antibody

Monoclonal antibodies (MoAbs) that recognize specific tumor antigens act through different mechanisms, including cell-mediated cytotoxicity, complement-dependent cytotoxicity, phagocytosis, apoptosis, and interference with specific signaling pathways [216,217].

CD20 is expressed on the surface of mature B-lymphocytes, but its expression is lost as B-cells differentiate into plasma cells. The expression of CD20 is maintained on the surface of lymphocytes of B-cell non-Hodgkin lymphomas. CD20 is part of a multimeric cell-surface complex that plays a role in regulating Ca^2+^ transport across the plasma membrane [218].

Rituximab belongs to the first generation of anti-CD20 MoAbs, is a murine-human chimeric antibody, and is used in B-cell malignancies. In contrast, both obinutuzumab and ofatumumab are classified as second-generation MoAbs against CD20, and they are either humanized or fully human. Importantly, these second-generation MoAbs exhibit lower immunogenicity and greater effectiveness in triggering apoptosis in B cells when compared to rituximab. The most prevalent cardiotoxicities of anti-CD20 MoAbs include infusion-related reactions, hypertension or hypotension, arrhythmias, and supraventricular tachycardia [154]. Maintaining Ca^2+^ homeostasis in the ER is crucial for supporting ER functions, such as protein folding and chaperone activity. The disturbance of this homeostasis, achieved through Ca^2+^ store depletion, triggers stress coping responses within the ER, including the unfolded protein response. Obinutuzumab has been observed to activate a Ca^2+^ response similar to that induced by rutuximab, involving the mobilization of ER Ca^2+^ stores and capacitive Ca^2+^ influx. This increase in Ca^2+^ plays a crucial role in inducing cell death [219,220].

Daratumumab and isatuximab, anti-CD38 MoAbs, are employed in multiple myeloma treatment due to the high expression of CD38 on neoplastic plasma cells. They are often used in conjunction with immunomodulatory drugs, proteasome inhibitors, and steroids. The most common cardiotoxic adverse events of anti-CD38 MoAbs include reactions related to infusion, hypertension, and arrhythmias such as atrial fibrillation and tachycardia [221].

Brentuximab vedotin consists of an anti-CD30 antibody conjugated by a protease-cleavable linker to a cytotoxic agent, auristatin E, that disrupts microtubules. The primary mode of action involves delivering auristatin E specifically to cells expressing CD30. Secondary mechanisms encompass phagocytosis dependent on antibodies, and immunogenic cell death [221,222].

## 10. Chimeric Antigen Receptor (CAR-T)

Chimeric antigen receptor T cells (CAR-T) are genetically engineered T-cells programmed to induce a cytotoxic immune response. CD19-directed CAR-T are used in relapsed or refractory B-cell lymphomas and acute lymphoblastic leukemia [223], whereas B-cell maturation antigen (BCMA)-directed CAR-T are used in multiple myeloma [224].

CAR-T therapy demonstrated associations with diverse cardiovascular and pulmonary adverse events, such as tachyarrhythmia, cardiomyopathy, pericardial and pleural disorders, and venous thromboembolism [225,226,227,228,229,230]. Moreover, tachyarrhythmias and thromboembolism were more commonly reported with axicabtagene-ciloleucel compared to tisagenlecleucel [63]. In addition to reported QT prolongation and arrhythmias, such as atrial fibrillation [231], patients with pre-existing cardiac structural abnormalities exhibited a higher incidence of cardiac troponin elevation, which was linked to subsequent cardiovascular events [226]. The primary adverse effect of CAR T-cell therapy is cytokine release syndrome (CRS), which clinically manifests in cardiovascular symptoms such as tachycardia, hypotension, troponin elevation, reduced LVEF, pulmonary edema, and cardiogenic shock [232]. CRS consists of elevated levels of inflammatory cytokines released by activated CAR T-cells and other immune cells, notably macrophages. CRS manifests as a systemic inflammatory response, impacting various organs with varying degrees of severity, ranging from mild to life-threatening conditions. These critical conditions may include cardiac dysfunction, adult respiratory distress syndrome, neurologic toxicity, coagulopathy, liver failure, and renal failure [232]. IL-6 has been implicated as a key mediator of this systemic inflammatory response, and recent findings also highlight activated endothelial cells as a crucial contributor to IL-6 production, playing a substantial modulatory role in the severity of CRS [233]. The overall approach to managing CRS involves the administration of tocilizumab, an IL-6 antagonist, particularly in severe cases, to reduce the risk of cardiotoxicity [234]. IL-6 can alter myocardial hemodynamics, resulting in a transient reduction of mean arterial pressure and left ventricle end systolic pressure. It induces negative cardiac inotropism. Furthermore, IL-6 increases phosphorylation of STAT3 and ERK1/2 (extracellular signal-regulated kinases) within minutes of exposure, leading to de novo synthesis of iNOS and subsequent NO production. iNO contributes to decreased cardiac contractility after 2 h of incubation with IL-6. These effects are abrogated with JAK2 inhibition. Proinflammatory cytokines like IL6 can exert negative inotropic and cytotoxic effects on cardiomyocytes [235,236].

Tocilizumab is a recombinant human monoclonal antibody that specifically binds soluble and membrane-bound IL-6 receptors (IL-6R), disrupting both classic and trans-signaling. Tocilizumab interferes with the cytokine feedback loop, blocks the inflammatory response, and decreases circulating levels of IL-6. Primarily indicated for juvenile idiopathic arthritis and rheumatoid arthritis, tocilizumab’s efficacy in managing severe CRS was also demonstrated in landmark studies that investigated CAR T-cell therapy [237,238].

In different clinical trials, ZUMA-1 [239], JULIET [240], ELIANA [241], CRS occurs in 60–70% of patients. Bachy et al. [242] compared two treatments with CAR T-cells, tisagenlecleucel and axicabtagene ciloleucel, in relapsed or refractory diffuse large B-cell lymphoma in adults and showed that axicabtagene ciloleucel had a significantly higher frequency of grade 1–2 CRS compared to tisagenlecleucel, although no significant difference was noted for grade ≥ 3. Concerning immune effector cell-associated neurotoxicity syndrome, there was a significantly higher occurrence of axicabtagene ciloleucel in comparison to tisagenlecleucel. In ZUMA-1 trial [239], tisagenlecleucel showed a lower incidence of grade ≥ 3 of immune effector cell-associated neurotoxicity syndrome at 12%, compared to axicabtagene ciloleucel, which had a higher incidence of 31% in the JULIET trial [240].

CAR T-cell therapy shows promise against hematological malignancies but faces challenges due to toxicity. Ongoing efforts aim to understand and improve safety, and recognizing and managing cardiotoxicity becomes crucial. Strategies for prevention and management are crucial, emphasizing the need for a best practice approach with systematic assessment involving cardiovascular symptoms, cardiac biomarkers such as cardiac troponin and brain natriuretic peptide, and imaging-based indices of cardiac function.

## 11. Bispecific Antibodies

Bispecific antibodies (BsAbs) are engineered molecules with different fragment antigen-binding (FABs) capable of binding to different antigens. If one FAB recognizes CD3, the BsAbs can attract T-lymphocytes in proximity to cells expressing the antigen recognized by the other FAB and elicit cytotoxicity [243].

In pretreated multiple myeloma, the BsAbs teclistamab, elranatamab and alnuctamab have shown efficacy in binding CD3 and BCMA, as well as the BsAbs talquetamab binding CD3 and GPRC5D [55,56]. BMCA is a B-cell maturation antigen, also known as tumor necrosis factor receptor superfamily member 17, overexpressed by multiple myeloma cells. Similarly, GPRC5D is overexpressed in multiple myeloma cells, but its function is not fully characterized. Adverse effects during elranatamab treatment include CRS, along with injection site reactions, upper respiratory tract infections, musculoskeletal pain, and pneumonia. Furthermore, the drug may induce grade 3 to 4 laboratory abnormalities, characterized by reduced lymphocyte, neutrophil, hemoglobin, white blood cell, and platelet counts [56]. In the case of CRS linked to BsAbs, it commonly manifests during the initial infusion. Similarly, teclistamab, alnuctamab and talquetamab can induce CRS with consequent cardiovascular symptoms [53,60].

Blinatumomab is a CD19/CD3 BsAb used for the treatment of acute lymphoblastic leukemia. Tachycardia and heart failure occur with a frequency below 1% in patients treated with blinatumomab [49]. Significant adverse effects of Blinatumomab encompass myelosuppression, CRS, and neurotoxicity.

The CD3/CD20 BsAbs mosunetuzumab, glofitamab, odronextamab, and epcoritamab have shown efficacy in refractory B-cell malignancies expressing CD20: Glofitamab and odronextamab have been studied in refractory diffuse large cell lymphoma, and mosunetuzumab in refractory follicular lymphoma. CRS is a frequent adverse event associated with the use of CD3/CD20 BsAbs [58,244,245,246]. Common CRS symptoms include pyrexia, hypotension, chills, headache, tachycardia, and hypoxia [247]. In addition, the study by Budde et al., revealed that CRS events linked to mosunetuzumab were mostly of low grade and confined to the initial treatment cycle [244].

Cardiac events associated with CAR-T cells are reasonably common, generally short-lived, reversible, and mainly occur in the setting of CRS. For patients at higher risk of cardiovascular adverse events who develop CRS, it is suggested to consider earlier administration of tocilizumab, aiming to reduce the severity of CRS. Data suggest that the risk of cardiac events with CRS increased 1.7-fold with each 12-h delay in tocilizumab administration. Patients with grade ≥2 CRS can be placed on telemetry for monitoring arrhythmias. An echocardiogram can be repeated in patients with a known history of reduced EF, cardiomyopathy, or pulmonary hypertension, as well as those with new symptoms of HF or hypotension. If baseline troponin and natriuretic peptide levels from patients are available, repeating biomarkers at the time of CRS may help guide the workup and monitoring [248].

## 12. Conclusions

Advancements in chemotherapy have led to better outcomes for cancer patients, resulting in a growing number of individuals living with a history of cancer. Many of these newer drugs have both short-term and long-term effects on the cardiovascular system. Understanding how these drugs interact with the heart and blood vessels is essential for predicting, detecting, and managing chemotherapy-related heart issues. Patients with pre-existing cardiovascular disorders face an elevated risk of drug-induced cardiotoxicity.

Consequently, there is an urgent demand for strategies to recognize and address cardiotoxicity during cancer therapy. This awareness has led to the emergence of a multidisciplinary field known as cardio-oncology. The main objectives of the rapidly expanding discipline of cardio-oncology include gaining a deeper understanding of the pathophysiology of cancer therapy-associated cardiotoxicity and offering early prediction, detection, management, and treatment of cardiac complications in patients with or survivors of cancer. These patients require pre-evaluation and ongoing cardiovascular risk monitoring after therapy exposure. It is crucial to follow recommended guidelines for continuous cardiotoxicity monitoring throughout treatments, incorporating both parameters like LVEF, CK-MB, cTn-I, natriuretic peptide, and ECG and echocardiogram.

While extensive efforts have been directed towards comprehending the mechanism of chemotherapy-induced cardiomyopathy, a significant challenge in the field is the absence of a suitable model system for conducting preclinical studies to accurately predict cardiotoxicity in humans.

## Figures and Tables

**Figure 1 jcm-13-01574-f001:**
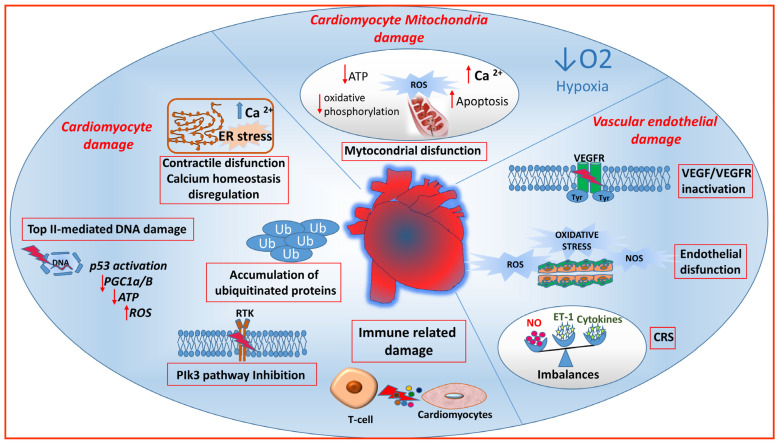
Molecular mechanisms of anticancer drug-mediated cardio-vascular toxicity. Chemotherapeutic agents and targeted treatments can induce cardiovascular toxicity by affecting both endothelial cells and cardiomyocytes. In vascular endothelial cells, the inhibition of the VEGF pathway can result in hypertension, altered vascular permeability, bleeding, and thrombosis. In endothelial cells, oxygen reactive species (ROS) can disrupt the equilibrium between vasoconstricting agents, including endothelin (ET-1), and vasodilating agents such as nitric oxide (NO) produced by nitric oxide synthase (NOS). In the cytokine release syndrome (CRS), vasoactive soluble factors cause vasodilation and hypotension. The impact of antineoplastic drugs on cardiomyocytes may interest the mitochondria and could be linked to hypoxia resulting from vascular damage or to a direct influence of the drugs on the mitochondria, especially on the respiratory chain. This can result in ATP depletion, heightened ROS, calcium ion release (Ca^2+^), and apoptosis. Additionally, drugs can induce damage to cardiomyocytes by causing endoplasmic reticulum (ER) stress, DNA damage due to topoisomerase 2β (Top II) inhibition, accumulation of ubiquitinated (Ub) proteins, and interference with intracellular pathways, including PI3K, through receptor tyrosine kinase (RTK) inhibition. Top II caused cardiomyocyte death by p53 activation leading to ROS generation and reduction of ATP and peroxisome proliferator–activated receptor-γ coactivator 1-α and 1-β (PGC1α/β). Furthermore, immune checkpoint inhibitors can stimulate T-lymphocytes to recognize self-antigens, leading to myocarditis.

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
