# Peer review of "Cardiovascular Toxicity of Antineoplastic Treatments in Hematological Diseases: Focus on Molecular Mechanisms to Improve Therapeutic Management"

_jcm, 2024, doi:10.3390/jcm13061574_

Round 1

Reviewer 1 Report

Comments and Suggestions for Authors

Barachini and Petrini, in the paper entitled "Cardiovascular Toxicity of Antineoplastic Treatments in 2 Hematological Diseases: Focus on Molecular Mechanisms to 3 Improve Therapeutic Management ", review the cardiac and vascular toxicity of antineoplastic drugs in hematological disorders, providing insights into the molecular physiopathology of cancer therapy-associated cardiotoxicity. 

The review is well structured and reviews the most important groups of drugs used in the treatment of haematological cancer. However, in my opinion, there are aspects that need to be corrected or expanded.

Introduction: it has an adequate extension and indicates all the relevant aspects necessary for the understanding of the work.

However, the following changes/revisions are suggested:           

·         The order in which the topic is presented is not appropriate. It should start with the most general and end with the most specific. It is suggested to reverse the order and also the order of the table and the figure. Lines 79-107 are general; lines 61-77 are more specific.

·         Citations in superscripts should be separated by commas.

·         Line 40: delete “and”.

Sections with different pharmacological groups of anti-tumour drugs.

They should have a separate individual introductory numbering and start with "2.”

Within each section, again go from the general to the specific aspects. Not all sections begin with a presentation of the mechanism of action of the pharmacological group. This aspect should be standardised across all groups.

Citations in superscripts should be separated by commas.

Lines 125, 145, 245, 276, 346: Words are incorrectly hyphenated at the end of the line.

In general, check abbreviations and order of abbreviations. They are not used correctly. Line 155 (P53 not correct), line 193 (Doxorubicin not abbreviated), line 598 (CRS used before explanation of the abbreviation (line 600-601)

Line 445: Change “inhibotors” for “inhibitors”.

Line 488: Change “Have” for “have”.

Lines 642-645. Already mentioned in the previous section. Eliminate repetitions.

Section "Alkylating and Platinum-Based Agents" and "Chimeric antigen receptor (CAR-T)".  Please, include preclinical studies on mechanisms.

Section "BCL2 inhibitors". Please, include clinical studies.

Comments on the Quality of English Language

In general, the writing of the manuscript is very difficult to follow, the expressions used are complex. It is important to revise the writing of the manuscript to make it easier to read.

Author Response

Dear Reviewer 1,

Thank you for taking the time to review my manuscript and for providing invaluable feedback. I am grateful for your positive comments on the overall organization and content of the review.

I would like to express my sincere gratitude for your comments on my manuscript. Your feedback has been helpful in enhancing the quality and relevance of the manuscript.

I hope that you find the revised version to be satisfactory and look forward to any further feedback you may have.

Reviewer 1

Barachini and Petrini, in the paper entitled "Cardiovascular Toxicity of Antineoplastic Treatments in Hematological Diseases: Focus on Molecular Mechanisms to improve Therapeutic Management ", review the cardiac and vascular toxicity of antineoplastic drugs in hematological disorders, providing insights into the molecular physiopathology of cancer therapy-associated cardiotoxicity. 

The review is well structured and reviews the most important groups of drugs used in the treatment of haematological cancer. However, in my opinion, there are aspects that need to be corrected or expanded.

Introduction: it has an adequate extension and indicates all the relevant aspects necessary for the understanding of the work.

However, the following changes/revisions are suggested:           

  • The order in which the topic is presented is not appropriate. It should start with the most general and end with the most specific. It is suggested to reverse the order and also the order of the table and the figure. Lines 79-107 are general; lines 61-77 are more specific. I have started with the most general information and concluded with the most specific details. Additionally, I have reversed the order of both the table and the figure.

           Citations in superscripts should be separated by commas. The commas are added.

          Line 40: delete “and”. I have done

Sections with different pharmacological groups of anti-tumour drugs.

They should have a separate individual introductory numbering and start with "2.” I have done

Within each section, again go from the general to the specific aspects. Not all sections begin with a presentation of the mechanism of action of the pharmacological group. This aspect should be standardised across all groups. The mechanisms of action of drugs are added where they are lacking.

Citations in superscripts should be separated by commas. The commas are added.

Lines 125, 145, 245, 276, 346: Words are incorrectly hyphenated at the end of the line. I appreciate your observation regarding the hyphenation issue but this process is independent of my actions as the author and is a task handled by the editorial process.

In general, check abbreviations and order of abbreviations. They are not used correctly. Line 155 (P53 not correct), line 193 (Doxorubicin not abbreviated), line 598 (CRS used before explanation of the abbreviation (line 600-601) I have corrected and reviewed all the abbreviations.

Line 445: Change “inhibotors” for “inhibitors”. I have done

Line 488: Change “Have” for “have”. I have done

Lines 642-645. Already mentioned in the previous section. Eliminate repetitions. I have done

Section "Alkylating and Platinum-Based Agents" and "Chimeric antigen receptor (CAR-T)".  Please, include preclinical studies on mechanisms. I have done the preclinical studies

Section "BCL2 inhibitors". Please, include clinical studies.I have added the clinical trials to the section.

Comments on the Quality of English Language

In general, the writing of the manuscript is very difficult to follow, the expressions used are complex. It is important to revise the writing of the manuscript to make it easier to read. I have reviewed the entire manuscript and clarified the parts that were difficult to understand.

Reviewer 2 Report

Comments and Suggestions for Authors

Comments:

In this review, the author delves into the complex landscape of cardiovascular toxicity associated with antineoplastic treatments in hematological diseases.

The introduction provides a comprehensive overview of the cardiovascular toxicities associated with antineoplastic treatments.

Table 1:

The description "DNA synthesis inhibitor/ DNA intercalating agent and DNA synthesis inhibitor" appears to be a redundancy. The term "DNA intercalating agent" typically refers to a substance that can insert itself between DNA base pairs, disrupting the DNA structure and inhibiting processes like replication or transcription. If it's intended to convey a dual mechanism, it should be clarified, or the repetition removed for clarity.

Provide more specific details about the adverse event. Instead of "Myocardiocytes damage," consider specifying the nature or consequences of the damage. "Myocardiocytes" is not commonly used, and "cardiomyocytes" is the more accepted and widely recognized term for the muscle cells of the heart. Therefore, it would be appropriate to use "cardiomyocytes" in the manuscript.

Consider adding additional relevant information such as the conditions or diseases for which a drug is used.

For the adverse events, it would be beneficial to include information on the incidence rates or prevalence of the mentioned cardiovascular complications, if available. Providing such details could give readers a clearer understanding of the risk associated with drug use.

Instead of a general term like "arrhythmias, conductivity dysfunction," specify the types of arrhythmias and conductivity dysfunctions that may occur. Be more detailed for all adverse events to give a comprehensive understanding.

For drugs please complete at mechanisms of action with Inhibition / stimulation.

Please explain in the legend of the table all the abbreviations used for a better understanding of the

reader.

The review effectively describes the molecular physiopathology of treatment-related adverse events, particularly focusing on anthracyclines. The detailed explanation of mechanisms involving topoisomerases, mitochondrial DNA binding, oxidative stress, and pathways leading to heart dysfunction is commendable. Please consider incorporating visual aids such as figures to illustrate complex pathways and enhance reader understanding. Visual representations can improve the accessibility of detailed molecular processes.

The detailed pathophysiology of cyclophosphamide-induced hypertension, including its impact on ATP production, fatty acid accumulation, and oxidative stress, is well-articulated. The inclusion of specific pathways (TRL-4, NF-kB, TGF-beta, TNF-alpha) adds complexity but may require some descriptions for a broader audience.

The mechanisms of action of BCR-ABL inhibitors are mentioned, but some details could be expanded for better clarity. For instance, explain the downstream signaling pathways affected by BCR-ABL inhibition.

Please explain how BTK inhibition affects specific pathways and molecular processes in the context of cardiac events.

The mechanisms underlying the cardiotoxic effects of ruxolitinib are briefly discussed. It would be beneficial to delve deeper into the specific pathways affected by JAK inhibition, providing a more comprehensive understanding of the potential cardiovascular implications.

The review mentions that "cardiotoxicity associated with ruxolitinib is not fully understood." Provide the latest evidence or studies that have investigated or shed light on this aspect to support the statement.

The mention of "late-onset cardiovascular diseases" lacks specificity. Clarify and specify the types of cardiovascular diseases or events associated with ruxolitinib, if available.

The mechanisms described, such as the inhibition of JAK/STAT signaling in adipose tissue, need more in-depth explanation. How does this inhibition specifically contribute to weight gain and interfere with growth hormone and leptin pathways?

The case of pulmonary hypertension with left ventricular dysfunction is briefly mentioned. Include more details about this case, such as patient characteristics, duration of treatment, and any other relevant clinical information.

While the PI3K/Akt/mTOR pathway is briefly mentioned, the manuscript could benefit from a more in-depth discussion of the mechanisms of action of PI3K inhibitors. Provide insights into how these inhibitors function at the molecular level.

The description of idelalisib and copanlisib is somewhat brief. Provide more details on their mechanisms of action, specificity, and any unique features that distinguish them from each other.

The manuscript mentions the CHRONOS-1 study briefly but lacks details on the study design, patient characteristics, and key findings. Include relevant data to support the safety and efficacy claims.

The reference to the study by AlAsmari et al. is made, but the manuscript lacks a critical assessment of the study design, sample size, and relevance to human populations. Include a brief critique of the mentioned study to evaluate its scientific validity.

Provide specific information on Venetoclax ` s organ toxicity, if available.

The description of how HDAC inhibitors work is quite basic. Elaborate on the mechanism of action, providing insights into how histone deacetylase inactivation affects P21 levels, cell cycle arrest, and apoptosis. Provide a more detailed molecular perspective.

The manuscript mentions sudden death cases from "torsades de pointes" in early clinical trials but attributes them to pre-existing cardiac comorbidities. Provide more comprehensive details on these cases, including patient profiles, trial conditions, and any efforts made to address the observed adverse outcomes.

While mentioning that the mechanisms of HDAC inhibitor-induced effects on cardiac repolarization are not well elucidated, the manuscript could benefit from a more critical analysis of current hypotheses and research gaps in understanding these mechanisms.

The manuscript mentions proteasome-mediated protein degradation and endothelial injury as potential mechanisms of cardiotoxicity but lacks depth in explaining these processes. Elaborate on how these mechanisms may lead to cardiotoxic events.

When comparing thalidomide, lenalidomide, and pomalidomide, provide a more comprehensive analysis of their individual cardiotoxic profiles. Specify if there are differences in the reported adverse events and the strength of evidence supporting these differences.

Be cautious when interpreting data from FAERS, as it relies on voluntary reporting and may not establish causation. Discuss limitations and potential biases associated with FAERS data.

The influence of Carfilzomib on vascular smooth muscle cells is briefly touched upon. Expand on this aspect, explaining how it exacerbates the vulnerability of atherosclerotic plaques and its potential clinical implications.

Please provide insights into how targeting CD20 affects calcium regulation and its relevance to cardiotoxicity.

When discussing brentuximab vedotin, delve deeper into the off-target effects of coupled cytotoxic agents, specifically auristatin E. Explain how these effects contribute to cardiotoxicity and discuss their clinical significance.

While tocilizumab is mentioned as a key player in CRS management, the manuscript should discuss the rationale and mechanisms behind using interleukin-6 antagonists. Provide insights into how this intervention mitigates cardiotoxicity and the evidence supporting its efficacy.

The manuscript introduces cytokine release syndrome (CRS) but does not delve into the nuances of its management or the specific strategies employed in the context of BsAbs. A more detailed discussion on how to mitigate CRS, especially its cardiovascular manifestations, would add value.

Conlusions

While highlighting the risk for patients with pre-existing cardiovascular disorders is relevant, the conclusion tends to overemphasize this aspect. Provide a more balanced discussion, considering the broader population of cancer patients.

The call for "strategies to recognize and address cardiotoxicity" is vague. Concrete recommendations or examples of ongoing research and interventions in the field of cardio-oncology would make the conclusion more impactful.

The conclusion acknowledges a significant challenge in the field but doesn't critically evaluate potential solutions or advancements. Providing insights into ongoing efforts to address the challenge would enhance the depth of the conclusion.

Please highlight the limitations of the study.

Comments on the Quality of English Language

Minor editing of English language required

Author Response

Dear Reviewer 2,

Thank you for taking the time to review my manuscript and for providing invaluable feedback.

I would like to express my sincere gratitude for your comments on my manuscript. Your feedback has been helpful in enhancing the quality and relevance of the manuscript.

I hope that you find the revised version to be satisfactory and look forward to any further feedback you may have.

Reviewer 2

Comments:

In this review, the author delves into the complex landscape of cardiovascular toxicity associated with antineoplastic treatments in hematological diseases.

The introduction provides a comprehensive overview of the cardiovascular toxicities associated with antineoplastic treatments.

Table 1:

The description "DNA synthesis inhibitor/ DNA intercalating agent and DNA synthesis inhibitor" appears to be a redundancy. The term "DNA intercalating agent" typically refers to a substance that can insert itself between DNA base pairs, disrupting the DNA structure and inhibiting processes like replication or transcription. If it's intended to convey a dual mechanism, it should be clarified, or the repetition removed for clarity. I removed the repetition for clarity.

Provide more specific details about the adverse event. Instead of "Myocardiocytes damage," consider specifying the nature or consequences of the damage. "Myocardiocytes" is not commonly used, and "cardiomyocytes" is the more accepted and widely recognized term for the muscle cells of the heart. Therefore, it would be appropriate to use "cardiomyocytes" in the manuscript. I have added more specific details about the adverse event where it was lacking and I correct in the text myocardiocytes with cardiomyocytes.

Consider adding additional relevant information such as the conditions or diseases for which a drug is used. I have done

For the adverse events, it would be beneficial to include information on the incidence rates or prevalence of the mentioned cardiovascular complications, if available. Providing such details could give readers a clearer understanding of the risk associated with drug use. Additional specific details have been added where they were previously lacking.

Instead of a general term like "arrhythmias, conductivity dysfunction," specify the types of arrhythmias and conductivity dysfunctions that may occur. Be more detailed for all adverse events to give a comprehensive understanding. I have done

For drugs please complete at mechanisms of action with Inhibition / stimulation.  I have done in the text

Please explain in the legend of the table all the abbreviations used for a better understanding of the reader. All the abbreviations used in the table are explained.

The review effectively describes the molecular physiopathology of treatment-related adverse events, particularly focusing on anthracyclines. The detailed explanation of mechanisms involving topoisomerases, mitochondrial DNA binding, oxidative stress, and pathways leading to heart dysfunction is commendable. Please consider incorporating visual aids such as figures to illustrate complex pathways and enhance reader understanding. Visual representations can improve the accessibility of detailed molecular processes. The figure has been improved by adding more information on the signaling pathways involved, building upon the existing figure.

The detailed pathophysiology of cyclophosphamide-induced hypertension, including its impact on ATP production, fatty acid accumulation, and oxidative stress, is well-articulated. The inclusion of specific pathways (TRL-4, NF-kB, TGF-beta, TNF-alpha) adds complexity but may require some descriptions for a broader audience. To enhance understanding, these specific pathways have been further elucidated with added descriptions and references.

The mechanisms of action of BCR-ABL inhibitors are mentioned, but some details could be expanded for better clarity. For instance, explain the downstream signaling pathways affected by BCR-ABL inhibition. Downstream signaling pathways affected by BCR-ABL inhibition are added.

Please explain how BTK inhibition affects specific pathways and molecular processes in the context of cardiac events. I have done

The mechanisms underlying the cardiotoxic effects of ruxolitinib are briefly discussed. It would be beneficial to delve deeper into the specific pathways affected by JAK inhibition, providing a more comprehensive understanding of the potential cardiovascular implications.  I have provided a clearer explanation of the mechanism and delved deeper into this specific signaling pathway affected by JAK inhibition, aiming to provide a more comprehensive understanding of the potential cardiovascular implications of ruxolitinib's cardiotoxic effects.

1-The review mentions that "cardiotoxicity associated with ruxolitinib is not fully understood." Provide the latest evidence or studies that have investigated or shed light on this aspect to support the statement. 2-The mention of "late-onset cardiovascular diseases" lacks specificity. Clarify and specify the types of cardiovascular diseases or events associated with ruxolitinib, if available. 3-The mechanisms described, such as the inhibition of JAK/STAT signaling in adipose tissue, need more in-depth explanation. How does this inhibition specifically contribute to weight gain and interfere with growth hormone and leptin pathways? In the section on Janus Kinase inhibitors, I have addressed these 3 points by providing the latest evidence and studies that shed light on the cardiotoxicity associated with ruxolitinib. I have also clarified and specified the types of cardiovascular diseases or events linked to ruxolitinib, where available. Additionally, I have provided a more in-depth explanation of the mechanisms of JAK/STAT signaling in adipose tissue. More references have been added for better comprehension.

The case of pulmonary hypertension with left ventricular dysfunction is briefly mentioned. Include more details about this case, such as patient characteristics, duration of treatment, and any other relevant clinical information. I have added these additional clinicals details.

While the PI3K/Akt/mTOR pathway is briefly mentioned, the manuscript could benefit from a more in-depth discussion of the mechanisms of action of PI3K inhibitors. Provide insights into how these inhibitors function at the molecular level. I have added this concern by providing a more in-depth discussion of the mechanisms of action of PI3K inhibitors, offering insights into how these inhibitors function at the molecular level. One reference has been added.

The description of idelalisib and copanlisib is somewhat brief. Provide more details on their mechanisms of action, specificity, and any unique features that distinguish them from each other. I have addressed this by providing more detailed information on the mechanisms of action that distinguish idelalisib and copanlisib from each other. One reference has been added.

The manuscript mentions the CHRONOS-1 study briefly but lacks details on the study design, patient characteristics, and key findings. Include relevant data to support the safety and efficacy claims. I have done

The reference to the study by AlAsmari et al. is made, but the manuscript lacks a critical assessment of the study design, sample size, and relevance to human populations. Include a brief critique of the mentioned study to evaluate its scientific validity. I have addressed this concern by providing more details of the study by AlAsmari et al., including an assessment of the study design, sample size, and relevance to human populations.

Provide specific information on Venetoclax ` s organ toxicity, if available. I have addressed this by adding specific information on Venetoclax's organ toxicities, specifically highlighting the occurrences of acute kidney injury and tumor lysis syndrome.

The description of how HDAC inhibitors work is quite basic. Elaborate on the mechanism of action, providing insights into how histone deacetylase inactivation affects P21 levels, cell cycle arrest, and apoptosis. Provide a more detailed molecular perspective. I have expanded the paragraph with more details on the mechanism of action of HDAC inhibitors, providing insights into how histone deacetylase inactivation influences p21 levels, cell cycle arrest, and apoptosis. More references have been added.

The manuscript mentions sudden death cases from "torsades de pointes" in early clinical trials but attributes them to pre-existing cardiac comorbidities. Provide more comprehensive details on these cases, including patient profiles, trial conditions, and any efforts made to address the observed adverse outcomes. I have expanded the paragraph with more details.

While mentioning that the mechanisms of HDAC inhibitor-induced effects on cardiac repolarization are not well elucidated, the manuscript could benefit from a more critical analysis of current hypotheses and research gaps in understanding these mechanisms. I have expanded the paragraph with more details on cardiac repolarization and on current hypothesis of this mechanism. One reference has been added.

The manuscript mentions proteasome-mediated protein degradation and endothelial injury as potential mechanisms of cardiotoxicity but lacks depth in explaining these processes. Elaborate on how these mechanisms may lead to cardiotoxic events. I have expanded the paragraph with more details on these processes.

When comparing thalidomide, lenalidomide, and pomalidomide, provide a more comprehensive analysis of their individual cardiotoxic profiles. Specify if there are differences in the reported adverse events and the strength of evidence supporting these differences. Cardiac toxicity has been reported for thalidomide, whereas for lenalidomide and pomalidomide, fewer cases have been described, and their cardiac safety profile is less relevant. As a result, the focus has been primarily on thalidomide due to the higher reported incidence of cardiac toxicity. 

Be cautious when interpreting data from FAERS, as it relies on voluntary reporting and may not establish causation. Discuss limitations and potential biases associated with FAERS data. I added this sentence in the text: “Although FAERS data rely on voluntary reporting and may not establish causation, it remains a standard reference for toxicity profiles”.

The influence of Carfilzomib on vascular smooth muscle cells is briefly touched upon. Expand on this aspect, explaining how it exacerbates the vulnerability of atherosclerotic plaques and its potential clinical implications.  I have expanded the paragraph with more details on these processes and I added one reference.

Please provide insights into how targeting CD20 affects calcium regulation and its relevance to cardiotoxicity. I have expanded the paragraph with more details on this process and I added two references.

When discussing brentuximab vedotin, delve deeper into the off-target effects of coupled cytotoxic agents, specifically auristatin E. Explain how these effects contribute to cardiotoxicity and discuss their clinical significance. I have added the off-target of coupled cytotoxic agents.

While tocilizumab is mentioned as a key player in CRS management, the manuscript should discuss the rationale and mechanisms behind using interleukin-6 antagonists. Provide insights into how this intervention mitigates cardiotoxicity and the evidence supporting its efficacy. I have explained the rationale and mechanisms behind using interleukin-6 antagonists. I added two references.

The manuscript introduces cytokine release syndrome (CRS) but does not delve into the nuances of its management or the specific strategies employed in the context of BsAbs. A more detailed discussion on how to mitigate CRS, especially its cardiovascular manifestations, would add value. I added more details on how mitigate the CRS. Two references were added. 

Conlusions

While highlighting the risk for patients with pre-existing cardiovascular disorders is relevant, the conclusion tends to overemphasize this aspect. Provide a more balanced discussion, considering the broader population of cancer patients.

The call for "strategies to recognize and address cardiotoxicity" is vague. Concrete recommendations or examples of ongoing research and interventions in the field of cardio-oncology would make the conclusion more impactful.

The conclusion acknowledges a significant challenge in the field but doesn't critically evaluate potential solutions or advancements. Providing insights into ongoing efforts to address the challenge would enhance the depth of the conclusion

Please highlight the limitations of the study

Minor editing of English language required I revised all the manuscript.

This is not a clinical study but a review. It is, of course, limited by retrospective data, but at the same time, it is quite comprehensive, covering a large number of hematological drugs. Our goal was to underline some well-known mechanisms of cardiotoxicity. We could suggest the introduction of a specific figure, like a cardio-oncologist, who can follow hematological patients from the beginning and assist the hematologist in using the best treatment with a lower cardiac toxicity profile.  Nevertheless, I added in the text: “These patients require pre-evaluation and ongoing cardiovascular risk monitoring after therapy exposure. It is crucial to follow recommended guidelines for continuous cardi-otoxicity monitoring throughout treatments, incorporating both parameters like LVEF, CK-MB, cTn-I, natriuretic peptide and ECG and echocardiogram”.

Round 2

Reviewer 1 Report

Comments and Suggestions for Authors

The authors have introduced, in the new version of the manuscript, all the suggested changes. This has improved the new version of the paper.